# NEON2: Finding Local Minima
# via First-Order Oracles

**Zeyuan Allen-Zhu**[*]
Microsoft Research AI
Redmond, WA 98052
zeyuan@csail.mit.edu

**Yuanzhi Li**[*]
Stanford University
Stanford, CA 94305
yuanzhil@stanford.edu

## Abstract

We propose a reduction for non-convex optimization that can (1) turn an stationary-point finding algorithm into an local-minimum finding one, and (2) replace the Hessian-vector product computations with only gradient computations. It works both in the stochastic and the deterministic settings, without hurting the algorithm's performance.

As applications, our reduction turns Natasha2 into a first-order method without hurting its theoretical performance. It also converts SGD, GD, SCSG, and SVRG into algorithms finding approximate local minima, outperforming some best known results.

## 1 Introduction

Nonconvex optimization has become increasingly popular due its ability to capture modern machine learning tasks in large scale. For instance, training neural nets corresponds to minimizing a function

$$f(x) = \frac{1}{n} \sum_{i=1}^{n} f_i(x)$$

over $x \in \mathbb{R}^d$ that is non-convex, where each training sample $i$ corresponds to one loss function $f_i(\cdot)$ in the summation. This average structure allows one to perform stochastic gradient descent (SGD) which uses a random $\nabla f_i(x)$ —corresponding to computing backpropagation once— to approximate $\nabla f(x)$ and performs descent updates.

Motivated by such large-scale machine learning applications, we wish to design faster first-order non-convex optimization methods that outperform the performance of gradient descent, both in the *online* and *offline* settings. In this paper, we say an algorithm is online if its complexity is independent of $n$ (so $n$ can be infinite), and offline otherwise. In recently years, researchers across different communities have gathered together to tackle this challenging question. By far, known theoretical approaches mostly fall into one of the following two categories.

**First-order methods for stationary points.** In analyzing first-order methods, we denote by gradient complexity $T$ the number of computations of $\nabla f_i(x)$. To achieve an $\varepsilon$-approximate stationary point —namely, a point $x$ with $\|\nabla f(x)\| \le \varepsilon$— it is a folklore that gradient descent (GD) is offline and needs $T \propto O\left(\frac{n}{\varepsilon^2}\right)$, while stochastic gradient decent (SGD) is online and needs $T \propto O\left(\frac{1}{\varepsilon^4}\right)$.

In recent years, the offline complexity has been improved to $T \propto O\left(\frac{n^{2/3}}{\varepsilon^2}\right)$ by the SVRG method [4, 24], and the online complexity has been improved to $T \propto O\left(\frac{1}{\varepsilon^{10/3}}\right)$ by the SCSG method [19]. Both of them rely on the so-called variance-reduction technique, originally discovered for convex problems [12, 17, 27, 29].

---

[*]Authors sorted in alphabetical order. We acknowledge a parallel work of Xu and Yang [31] (which appeared online a few days before us), and have adopted their algorithm name Neon and called our new algorithm Neon2. Our algorithms are very different from theirs, and give better theoretical performance. The full version of this paper can be found on https://arxiv.org/abs/1711.06673.

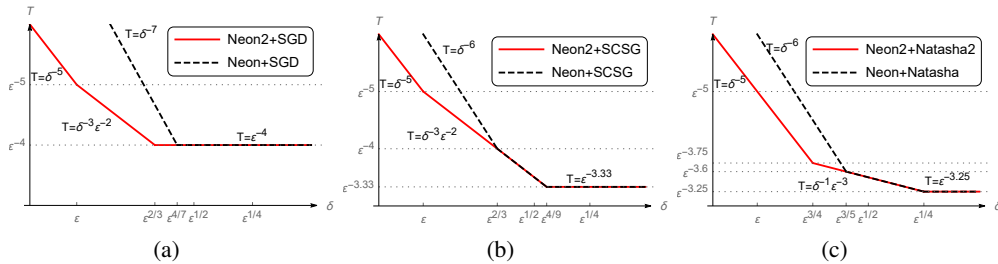

Figure 1: `Neon` vs `Neon2` for finding $(\varepsilon, \delta)$-approximate local minima. We emphasize that `Neon2` and `Neon` try to tackle the same problem, but are different algorithms.

Both algorithms `SVRG` and `SCSG` are only capable of finding approximate *stationary points*, which may not necessarily be approximate local minima and are arguably bad solutions for deep neural nets [10, 11, 15]. Thus,

*can we turn stationary-point finding algorithms into local-minimum finding ones?*

**Hessian-vector methods for local minima.** Using information about the Hessian, one can find $\varepsilon$-approximate local minima —namely, a point $x$ with $\|\nabla f(x)\| \le \varepsilon$ and also $\nabla^2 f(x) \succeq -\varepsilon^{1/C}\mathbf{I}$.[2] In 2006, Nesterov and Polyak [21] showed that one can find an $\varepsilon$-approximate in $O(\frac{1}{\varepsilon^{1.5}})$ iterations, but each iteration requires an (offline) computation as heavy as inverting the matrix $\nabla^2 f(x)$.

To fix this issue, researchers propose to study the so-called "Hessian-free" methods that, in addition to gradient computations, also compute Hessian-vector products. That is, instead of using the full matrix $\nabla^2 f_i(x)$ or $\nabla^2 f(x)$, these methods also compute $\nabla^2 f_i(x) \cdot v$ for indices $i$ and vectors $v$.[3] For Hessian-free methods, we denote by gradient complexity $T$ the number of computations of $\nabla f_i(x)$ plus that of $\nabla^2 f_i(x) \cdot v$. The hope of using Hessian-vector products is to improve the complexity $T$ as a function of $\varepsilon$.

Such improvement was first shown possible independently by [1, 8] for the offline setting, with complexity $T \propto \left( \frac{n}{\varepsilon^{1.5}} + \frac{n^{3/4}}{\varepsilon^{1.75}} \right)$ so is better than that of gradient descent. In the online setting, the first improvement was by `Natasha2` which gives complexity $T \propto \left( \frac{1}{\varepsilon^{3.25}} \right)$ [2].

Unfortunately, it is argued by some researchers that Hessian-vector products are not general enough and may not be as simple to implement as evaluating gradients [9]. Therefore,

*can we turn Hessian-free methods into first-order ones, without hurting their performance?*

## 1.1 From Hessian-Vector Products to First-Order Methods

Recall by definition of derivative we have

$$\nabla^2 f_i(x) \cdot v = \lim_{q \to 0} \left\{ \frac{\nabla f_i(x+qv) - \nabla f_i(x)}{q} \right\} \ .$$

Given any Hessian-free method, at least at a high level, can we replace every occurrence of $\nabla^2 f_i(x) \cdot v$ with $w = \frac{\nabla f_i(x+qv) - \nabla f_i(x)}{q}$ for some small $q > 0$?

Note the error introduced in this approximation is $\|\nabla^2 f_i(x) \cdot v - w\| \propto q\|v\|^2$. However, the original algorithm might not be stable to adversarial noise, thus, an (inverse) exponentially small $q$ might be required. One of our main contributions is to show how to implement these algorithms stably, so we can convert Hessian-free methods into first-order ones with an (inverse) *polynomially small* $q$.

In this paper, we demonstrate this idea by converting *negative-curvature-search (NC-search)* subroutines into first-order processes. NC-search is a key subroutine used in state-of-the-art Hessian-free methods that have rigorous proofs [1, 2, 8]. It solves the following simple task:

───────── **negative-curvature search (NC-search)** ─────────

given point $x_0$, decide if $\nabla^2 f(x_0) \succeq -\delta\mathbf{I}$ or find a unit vector $v$ such that $v^\top \nabla^2 f(x_0) v \le -\frac{\delta}{2}$.

───────────────────────────────────

[2] We say $\mathbf{A} \succeq -\delta\mathbf{I}$ if all the eigenvalues of $\mathbf{A}$ are no smaller than $-\delta$. In this high-level introduction, we focus only on the case when $\delta = \varepsilon^{1/C}$ for some constant $C$.

[3] Hessian-free methods are useful because when $f_i(\cdot)$ is explicitly given, computing its gradient is in the same complexity as computing its Hessian-vector product [23] [28], using backpropagation.

**Online Setting.** In the online setting, NC-search can be solved by Oja's algorithm [22] which costs $\widetilde{O}(1/\delta^2)$ computations of Hessian-vector products. This is first proved by Allen-Zhu and Li [7] and first applied to NC-search in Natasha2 [2]).

In this paper, we propose a method Neon2$^{\mathrm{online}}$ which solves the NC-search problem via only stochastic first-order updates. That is, starting from $x_1 = x_0 + \xi$ where $\xi$ is some random perturbation, we keep updating $x_{t+1} = x_t - \eta(\nabla f_i(x_t) - \nabla f_i(x_0))$. In the end, the vector $x_T - x_0$ gives us enough information about the negative curvature.

**Theorem 1** (informal). *Our* Neon2$^{\mathrm{online}}$ *algorithm solves NC-search using* $\widetilde{O}(1/\delta^2)$ *stochastic gradients, without Hessian-vector product computations.*

This complexity $\widetilde{O}(1/\delta^2)$ matches that of Oja's algorithm, and is information-theoretically optimal (up to log factors), see the lower bound in [7].

The independent work Neon by Xu and Yang [31] is actually the *first* recorded theoretical result that proposed this approach. However, Neon needs $\widetilde{O}(1/\delta^3)$ stochastic gradients, because it uses full gradient descent to find NC (on a sub-sampled objective) inspired by power method and [16]; instead, Neon2$^{\mathrm{online}}$ uses *stochastic gradients* and is based on the recent result of Oja's algorithm [7].

Plugging Neon2$^{\mathrm{online}}$ into Natasha2 [2], we achieve the following corollary (see Figure 1(c)):

**Theorem 2** (informal). Neon2$^{\mathrm{online}}$ *turns* Natasha2 *into a stochastic first-order method, without hurting its performance. That is, it finds an* $(\varepsilon, \delta)$*-approximate local minimum in* $T = \widetilde{O}\big(\frac{1}{\varepsilon^{3.25}} + \frac{1}{\varepsilon^3\delta} + \frac{1}{\delta^5}\big)$ *stochastic gradient computations, without Hessian-vector product computations.*

(We say $x$ is an $(\varepsilon, \delta)$-approximate local minimum if $\|\nabla f(x)\| \leq \varepsilon$ and $\nabla^2 f(x) \succeq -\delta\mathbf{I}$.)

**Offline Deterministic Setting.** There are a number of ways to solve the NC-search problem in the offline setting using Hessian-vector products. Most notably, power method uses $\widetilde{O}(n/\delta)$ computations of Hessian-vector products, and Lanscoz method [18] uses $\widetilde{O}(n/\sqrt{\delta})$ computations.

In this paper, we convert (a variant of) Lanscoz's method into a first-order one:

**Theorem 3** (informal). *Our* Neon2$^{\mathrm{det}}$ *algorithm solves NC-search using* $\widetilde{O}(1/\sqrt{\delta})$ *full gradients (or equivalently* $\widetilde{O}(n/\sqrt{\delta})$ *stochastic gradients).*

The independent work Neon [31] also applies to the offline setting, and needs $\widetilde{O}(1/\delta)$ full gradients. Their approach is inspired by [16], but our Neon2$^{\mathrm{det}}$ is based on Chebyshev approximation theory.

By putting Neon2$^{\mathrm{det}}$ and Neon2$^{\mathrm{finite}}$ into the CDHS method of Carmon et al. [8], we have

**Theorem 4** (informal). Neon2$^{\mathrm{det}}$ *turns* CDHS *into a first-order method without hurting its performance: it finds an* $(\varepsilon, \delta)$*-approximate local minimum in* $\widetilde{O}\big(\frac{1}{\varepsilon^{1.75}} + \frac{1}{\delta^{3.5}}\big)$ *full gradient computations.*

### 1.1.1 Offline Finite-Sum Setting

Recall one can also solve the NC-search problem in the offline setting by the (finite-sum) shift-and-invert [13] method, using $\widetilde{O}(n + n^{3/4}/\sqrt{\delta})$ computations of Hessian-vector products. We refer to this method as "finite-sum SI", and also convert it into a first-order method.

**Theorem 5** (informal). Neon2$^{\mathrm{finite}}$ *algorithm solves NC-search using* $\widetilde{O}(n + n^{3/4}/\sqrt{\delta})$ *stochastic gradients.*

Putting Neon2$^{\mathrm{finite}}$ into the (finite-sum version of) CDHS method [8], we have[4]

**Theorem 6** (informal). Neon2$^{\mathrm{finite}}$ *turns* CDHS *into a first-order method without hurting its performance: it finds an* $(\varepsilon, \delta)$*-approximate local minimum in* $T = \widetilde{O}\big(\frac{n}{\varepsilon^{1.5}} + \frac{n}{\delta^3} + \frac{n^{3/4}}{\varepsilon^{1.75}} + \frac{n^{3/4}}{\delta^{3.5}}\big)$ *stochastic gradient computations.*

*Remark* 1.1. All the cited works in Section 1.1 requires the objective to have (1) Lipschitz-continuous Hessian and (2) Lipschitz-continuous gradient. One can argue that (1) and (2) are both necessary for finding approximate local minima, but if only finding approximate stationary points, then only (2) is necessary. We shall formally discuss our assumptions in Section 2.

| | algorithm | gradient complexity $T$ | Hessian-vector products | variance bound | Lip. smooth | 2$^{\text{nd}}$-order smooth |
|---|---|---|---|---|---|---|
| stationary | SGD (folklore) | $O\left(\frac{1}{\varepsilon^4}\right)$ | no | needed | needed | no |
| local minima | perturbed SGD [14] | $\widetilde{O}\left(\frac{\mathrm{poly}(d)}{\varepsilon^4} + \frac{\mathrm{poly}(d)}{\delta^{16}}\right)$ | no | needed | needed | needed |
| | Neon+SGD [31] | $\widetilde{O}\left(\frac{1}{\varepsilon^4} + \frac{1}{\delta^7}\right)$ | no | needed | needed | needed |
| | Neon2+SGD | $\widetilde{O}\left(\frac{1}{\varepsilon^4} + \frac{1}{\varepsilon^2\delta^3} + \frac{1}{\delta^5}\right)$ | no | needed | needed | needed |
| stationary | SCSG [19] | $O\left(\frac{1}{\varepsilon^{10/3}}\right)$ | no | needed | needed | no |
| local minima | Neon+SCSG [31] | $O\left(\frac{1}{\varepsilon^{10/3}} + \frac{1}{\varepsilon^2\delta^3} + \frac{1}{\delta^6}\right)$ | no | needed | needed | needed |
| | Neon2+SCSG | $O\left(\frac{1}{\varepsilon^{10/3}} + \frac{1}{\varepsilon^2\delta^3} + \frac{1}{\delta^5}\right)$ | no | needed | needed | needed |
| local minima | Natasha2 [2] | $\widetilde{O}\left(\frac{1}{\varepsilon^{3.25}} + \frac{1}{\varepsilon^3\delta} + \frac{1}{\delta^5}\right)$ | needed | needed | needed | needed |
| | Neon+Natasha2 [31] | $\widetilde{O}\left(\frac{1}{\varepsilon^{3.25}} + \frac{1}{\varepsilon^3\delta} + \frac{1}{\delta^6}\right)$ | no | needed | needed | needed |
| | Neon2+Natasha2 | $\widetilde{O}\left(\frac{1}{\varepsilon^{3.25}} + \frac{1}{\varepsilon^3\delta} + \frac{1}{\delta^5}\right)$ | no | needed | needed | needed |
| | ↑ online methods ↑ | ↓ offline methods ↓ | | | | |
| stationary | GD (folklore [20]) | $O\left(\frac{n}{\varepsilon^2}\right)$ | no | no | needed | no |
| local minima | perturbed GD [16] | $\widetilde{O}\left(\frac{n}{\varepsilon^2} + \frac{n}{\delta^4}\right)$ | no | no | needed | needed |
| | Neon2+GD | $\widetilde{O}\left(\frac{n}{\varepsilon^2} + \frac{n}{\delta^{3.5}}\right)$ | no | no | needed | needed |
| stationary | SVRG [24] [4] | $O\left(\frac{n^{2/3}}{\varepsilon^2} + n\right)$ | no | no | needed | no |
| local minima | Reddi et al. [25] | $\widetilde{O}\left(\frac{n^{2/3}}{\varepsilon^2} + \frac{n}{\delta^3} + \frac{n^{3/4}}{\delta^{3.5}}\right)$ | needed | no | needed | needed |
| | Neon2+SVRG | | no | no | needed | needed |
| stationary | "convex until guilty" [9] | $\widetilde{O}\left(\frac{n}{\varepsilon^{1.75}}\right)$ | no | no | needed | needed |
| local minima | FastCubic [1] CDHS [8] | $\widetilde{O}\left(\frac{n}{\varepsilon^{1.5}} + \frac{n}{\delta^3} + \frac{n^{3/4}}{\varepsilon^{1.75}} + \frac{n^{3/4}}{\delta^{3.5}}\right)$ | needed | no | needed | needed |
| | Neon2+CDHS | $\widetilde{O}\left(\frac{n}{\varepsilon^{1.5}} + \frac{n}{\delta^3} + \frac{n^{3/4}}{\varepsilon^{1.75}} + \frac{n^{3/4}}{\delta^{3.5}}\right)$ | no | no | needed | needed |

Table 1: Complexity for finding $\|\nabla f(x)\| \leq \varepsilon$ and $\nabla^2 f(x) \succeq -\delta\mathbf{I}$. Following tradition, in these complexity bounds, we assume variance and smoothness parameters as constants, and only show the dependency on $n, d, \varepsilon$.

**Remark 1.** Variance bounds is needed for online methods (first half of the table).

**Remark 2.** Lipschitz smoothness is needed for finding even approximate stationary points.

**Remark 3.** Second-order Lipschitz smoothness is needed for finding approximate local minima.

## 1.2 From Stationary Points to Local Minima

Given any first-order method that finds stationary points (such as GD, SGD, SVRG or SCSG), we can hope for using the NC-search routine to identify whether or not its output $x$ satisfies $\nabla^2 f(x) \succeq -\delta\mathbf{I}$. If so, then automatically $x$ becomes an $(\varepsilon, \delta)$-approximate local minima so we can terminate. If not, we can go into its negative curvature direction to further decrease the objective.

In the independent work of Xu and Yang [31], they applied their Neon method for NC-search, and thus turned SGD and SCSG into first-order methods finding approximate local minima. In this paper, we use Neon2 instead. We show the following theorem:

**Theorem 7** (informal). *To find an $(\varepsilon, \delta)$-approximate local minima,*

---

**Algorithm 1** $\texttt{Neon2}_{\texttt{weak}}^{\texttt{online}}(f, x_0, \delta)$

---

1: $\eta \leftarrow \frac{\delta}{C_0^2 L^2 \log(100d)}$, $T \leftarrow \frac{C_0^2 \log(100d)}{\eta\delta}$,  $\diamond$ *for sufficiently large constant* $C_0$

2: $\xi \leftarrow \sigma \frac{\xi'}{\|\xi'\|_2}$ where $\xi' \sim \mathcal{N}(0, \mathbf{I})$.  $\diamond$ $\xi$ *is Gaussian random vector with norm* $\sigma := (100d)^{-3C_0} \frac{\eta^2\delta^3}{L_2}$

3: $x_1 \leftarrow x_0 + \xi$.

4: **for** $t \leftarrow 1$ **to** $T$ **do**

5: $\quad x_{t+1} \leftarrow x_t - \eta\left(\nabla f_i(x_t) - \nabla f_i(x_0)\right)$ where $i \in_R [n]$.

6: $\quad$ **if** $\|x_{t+1} - x_0\|_2 \geq r$ **then return** $v = \frac{x_s - x_0}{\|x_s - x_0\|_2}$ for a uniformly random $s \in [t]$.
$\diamond$ $r := (100d)^{C_0}\sigma$

7: **end for**

8: **return** $v = \bot$;

---

*(a)* $\texttt{Neon2+SGD}$ *needs* $T = \widetilde{O}\left(\frac{1}{\varepsilon^4} + \frac{1}{\varepsilon^2\delta^3} + \frac{1}{\delta^5}\right)$ *stochastic gradients;*

*(b)* $\texttt{Neon2+SCSG}$ *needs* $T = \widetilde{O}\left(\frac{1}{\varepsilon^{10/3}} + \frac{1}{\varepsilon^2\delta^3} + \frac{1}{\delta^5}\right)$ *stochastic gradients; and*

*(c)* $\texttt{Neon2+GD}$ *needs* $T = \widetilde{O}\left(\frac{n}{\varepsilon^2} + \frac{n}{\delta^{3.5}}\right)$ *(so* $\widetilde{O}\left(\frac{1}{\varepsilon^2} + \frac{1}{\delta^{3.5}}\right)$ *full gradients).*

*(d)* $\texttt{Neon2+SVRG}$ *needs* $T = \widetilde{O}\left(\frac{n^{2/3}}{\varepsilon^2} + \frac{n}{\delta^3} + \frac{n^{3/4}}{\delta^{3.5}}\right)$ *stochastic gradients.*

### 1.3 Roadmap

We introduce notions and formalize the problem in Section 2. We introduce $\texttt{Neon2}$ in the online, deterministic, and SVRG settings respectively in Section 3, Section 4 and Section 5. We apply $\texttt{Neon2}$ to SGD, GD, $\texttt{Natasha2}$, CDHS, SCSG and SVRG in Section 6. Most of the proofs are in the appendix.

## 2 Preliminaries

Throughout this paper, we denote by $\|\cdot\|$ the Euclidean norm. We use $i \in_R [n]$ to denote that $i$ is generated from $[n] = \{1, 2, \ldots, n\}$ uniformly at random. We denote by $\mathbb{I}[event]$ the indicator function of probabilistic events.

We denote by $\|\mathbf{A}\|_2$ the spectral norm of matrix $\mathbf{A}$. For symmetric matrices $\mathbf{A}$ and $\mathbf{B}$, we write $\mathbf{A} \succeq \mathbf{B}$ to indicate that $\mathbf{A} - \mathbf{B}$ is positive semidefinite (PSD). Therefore, $\mathbf{A} \succeq -\sigma\mathbf{I}$ if and only if all eigenvalues of $\mathbf{A}$ are no less than $-\sigma$. We denote by $\lambda_{\min}(\mathbf{A})$ and $\lambda_{\max}(\mathbf{A})$ the minimum and maximum eigenvalue of a symmetric matrix $\mathbf{A}$.

**Definition 2.1.** *For a function* $f \colon \mathbb{R}^d \to \mathbb{R}$,

- $f$ *is $L$-Lipschitz smooth (or $L$-smooth for short) if* $\forall x, y \in \mathbb{R}^d$, $\|\nabla f(x) - \nabla f(y)\| \leq L\|x - y\|$.
- $f$ *is second-order $L_2$-Lipschitz smooth (or $L_2$-second-order smooth for short) if*

$$\forall x, y \in \mathbb{R}^d, \|\nabla^2 f(x) - \nabla^2 f(y)\|_2 \leq L_2\|x - y\| .$$

### 2.1 Problem and Assumptions

Throughout the paper we study

$$\min_{x \in \mathbb{R}^d} \left\{ f(x) := \frac{1}{n} \sum_{i=1}^n f_i(x) \right\} \tag{2.1}$$

where both $f(\cdot)$ and each $f_i(\cdot)$ can be nonconvex. We wish to find $(\varepsilon, \delta)$-approximate local minima which are points $x$ satisfying

$$\|\nabla f(x)\| \leq \varepsilon \quad \text{and} \quad \nabla^2 f(x) \succeq -\delta\mathbf{I} .$$

We need the following three assumptions

- Each $f_i(x)$ is $L$-Lipschitz smooth.

- Each $f_i(x)$ is second-order $L_2$-Lipschitz smooth.

- Stochastic gradients have bounded variance: $\forall x \in \mathbb{R}^d$: $\quad \mathbb{E}_{i \in_R [n]} \|\nabla f(x) - \nabla f_i(x)\|^2 \leq \mathcal{V} .$

   (This assumption is needed only for *online* algorithms.)

**Algorithm 2** Neon2$^{\text{online}}(f, x_0, \delta, p)$  $\qquad\qquad\qquad\qquad\diamond$ *for boosting confidence of* Neon2$^{\text{online}}_{\text{weak}}$

---

**Input:** Function $f(x) = \frac{1}{n}\sum_{i=1}^{n} f_i(x)$, vector $x_0$, negative curvature $\delta > 0$, confidence $p \in (0, 1]$.
1: **for** $j = 1, 2, \cdots \Theta(\log 1/p)$ **do**  $\qquad\qquad\qquad\qquad\qquad\qquad\diamond$ *boost the confidence*
2: $\qquad v_j \leftarrow$ Neon2$^{\text{online}}_{\text{weak}}(f, x_0, \delta)$;
3: $\qquad$**if** $v_j \neq \perp$ **then**
4: $\qquad\qquad m \leftarrow \Theta(\frac{L^2 \log 1/p}{\delta^2})$, $v' \leftarrow \Theta(\frac{\delta}{L_2})v$.
5: $\qquad\qquad$Draw $i_1, \ldots, i_m \in_R [n]$.
6: $\qquad\qquad z_j = \frac{1}{m\|v'\|_2^2} \sum_{j=1}^{m} (v')^\top \left(\nabla f_{i_j}(x_0 + v') - \nabla f_{i_j}(x_0)\right)$
7: $\qquad\qquad$**if** $z_j \leq -3\delta/4$ **return** $v = v_j$
8: $\qquad$**end if**
9: **end for**
10: **return** $v = \perp$.

---

## 3 Neon2 in the Online Setting

We propose Neon2$^{\text{online}}$ as the online version of Neon2. It repeatedly invokes Neon2$^{\text{online}}_{\text{weak}}$ in Algorithm 1, whose goal is to solve the NC-search problem with confidence $2/3$ only; then Neon2$^{\text{online}}$ invokes Neon2$^{\text{online}}_{\text{weak}}$ repeatedly for $\log(1/p)$ times to boost the confidence to $1 - p$.

We prove the following theorem:

**Theorem 1** (Neon2$^{\text{online}}$). *Let $f(x) = \frac{1}{n}\sum_{i=1}^{n} f_i(x)$ where each $f_i$ is L-smooth and $L_2$-second-order smooth. For every point $x_0 \in \mathbb{R}^d$, every $\delta \in (0, L]$, every $p \in (0, 1)$, the output*

$$v = \texttt{Neon2}^{\text{online}}(f, x_0, \delta, p)$$

*satisfies that, with probability at least $1 - p$:*

*1. If $v = \perp$, then $\nabla^2 f(x_0) \succeq -\delta \mathbf{I}$.*

*2. If $v \neq \perp$, then $\|v\|_2 = 1$ and $v^\top \nabla^2 f(x_0) v \leq -\frac{\delta}{2}$.*

*Moreover, the total number of stochastic gradient evaluations $O\left(\frac{\log^2(d/p)L^2}{\delta^2}\right)$.*

The proof of Theorem 1 immediately follows from Lemma 3.1 and Lemma 3.2 below.

**Lemma 3.1** (Neon2$^{\text{online}}_{\text{weak}}$). *In the same setting as Theorem 1, the output $v = \texttt{Neon2}^{\text{online}}_{\text{weak}}(f, x_0, \delta)$ satisfies If $\lambda_{\min}(\nabla^2 f(x_0)) \leq -\delta$, then with probability at least $2/3$, $v \neq \perp$ and $v^\top \nabla^2 f(x_0) v \leq -\frac{3}{4}\delta$.*

*Proof sketch of Lemma 3.1.* We explain why Neon2$^{\text{online}}_{\text{weak}}$ works as follows. Starting from a randomly perturbed point $x_1 = x_0 + \xi$, it keeps updating $x_{t+1} \leftarrow x_t - \eta (\nabla f_i(x_t) - \nabla f_i(x_0))$ for some random index $i \in [n]$, and stops either when $T$ iterations are reached, or when $\|x_{t+1} - x_0\|_2 > r$. Therefore, we have $\|x_t - x_0\|_2 \leq r$ throughout the iterations, and thus can approximate $\nabla^2 f_i(x_0)(x_t - x_0)$ using $\nabla f_i(x_t) - \nabla f_i(x_0)$, up to error $O(r^2)$. This is a small term based on our choice of $r$.

Ignoring the error term, our updates look like $x_{t+1} - x_0 = \left(\mathbf{I} - \eta \nabla^2 f_i(x_0)\right)(x_t - x_0)$. This is exactly the same as Oja's algorithm [22] which is known to approximately compute the minimum eigenvector of $\nabla^2 f(x_0) = \frac{1}{n}\sum_{i=1}^{n} f_i(x_0)$. Using the recent optimal convergence analysis of Oja's algorithm [7], one can conclude that after $T_1 = \Theta\left(\frac{\log \frac{r}{\alpha}}{\eta\lambda}\right)$ iterations, where $\lambda = \max\{0, -\lambda_{\min}(\nabla^2 f(x_0))\}$, then we not only have that $\|x_{t+1} - x_0\|_2$ is blown up, but also it aligns well with the minimum eigenvector of $\nabla^2 f(x_0)$. In other words, if $\lambda \geq \delta$, then the algorithm must stop before $T$.

Finally, one has to carefully argue that the error does not blow up in this iterative process. We defer the proof details to Appendix B.3. $\qquad\qquad\qquad\qquad\qquad\qquad\qquad\qquad\square$

Our Lemma 3.2 below tells us we can verify if the output $v$ of Neon2$^{\text{online}}_{\text{weak}}$ is indeed correct (up to additive $\frac{\delta}{4}$), so we can boost the success probability to $1 - p$. For completeness' sake, we summarize this procedure as Neon2$^{\text{online}}$ in Algorithm 2.

**Lemma 3.2** (verification). *In the same setting as Theorem 1, let vectors $x, v \in \mathbb{R}^d$. If $i_1, \ldots, i_m \in_R$ $[n]$ and define*

$$z = \frac{1}{m} \sum_{j=1}^{m} v^\top (\nabla f_{i_j}(x + v) - \nabla f_{i_j}(x))$$

*Then, if $\|v\| \leq \frac{\delta}{8L_2}$ and $m = \Theta\left(\frac{L^2 \log 1/p}{\delta^2}\right)$, with probability at least $1 - p$,*

$$\left| \frac{z}{\|v\|_2^2} - \frac{v^\top \nabla^2 f(x) v}{\|v\|_2^2} \right| \leq \frac{\delta}{4} \quad .$$

The simple proof of Lemma 3.2 can be found in Section B.4.

## 4 Neon2 in the Deterministic Setting

---
**Algorithm 3** $\texttt{Neon2}^{\text{det}}(f, x_0, \delta, p)$

---
**Input:** A function $f$, vector $x_0$, negative curvature target $\delta > 0$, failure probability $p \in (0, 1]$.
1: $T \leftarrow \frac{C_1^2 \log(d/p)\sqrt{L}}{\sqrt{\delta}}$.                 $\diamond$ *for sufficiently large constant $C_1$.*
2: $\xi \leftarrow$ Gaussian random vector with norm $\sigma$;          $\diamond$ $\sigma := (d/p)^{-2C_1} \frac{\delta}{T^4 L_2}$
3: $x_1 \leftarrow x_0 + \xi$. $y_1 \leftarrow \xi, y_0 \leftarrow 0$
4: **for** $t \leftarrow 1$ **to** $T$ **do**
5:      $y_{t+1} = 2\mathcal{M}(y_t) - y_{t-1}$;       $\diamond$ $\mathcal{M}(y) := -\frac{1}{L} (\nabla f(x_0 + y) - \nabla f(x_0)) + \left(1 - \frac{3\delta}{4L}\right) y$
6:      $x_{t+1} = x_0 + y_{t+1} - \mathcal{M}(y_t)$.
7:      **if** $\|x_{t+1} - x_0\|_2 \geq r$ **then return** $\frac{x_{t+1} - x_0}{\|x_{t+1} - x_0\|_2}$.      $\diamond$ $r := (d/p)^{C_1} \sigma$
8: **end for**
9: **return** $\perp$.

---

We propose $\texttt{Neon2}^{\text{det}}$ formally in Algorithm 3 and prove:

**Theorem 3** ($\texttt{Neon2}^{\text{det}}$). *Let $f(x)$ be a function that is $L$-smooth and $L_2$-second-order smooth. For every point $x_0 \in \mathbb{R}^d$, every $\delta > 0$, every $p \in (0, 1]$, the output $v = \texttt{Neon2}^{\text{det}}(f, x_0, \delta, p)$ satisfies that, with probability at least $1 - p$:*

*1. If $v = \perp$, then $\nabla^2 f(x_0) \succeq -\delta \mathbf{I}$.*

*2. If $v \neq \perp$, then $\|v\|_2 = 1$ and $v^\top \nabla^2 f(x_0) v \leq -\frac{\delta}{2}$.*

*Moreover, the total number full gradient evaluations is $O\left(\frac{\log^2(d/p)\sqrt{L}}{\sqrt{\delta}}\right)$.*

*Proof sketch of Theorem 3.* We explain the high-level intuition of $\texttt{Neon2}^{\text{det}}$ and the proof of Theorem 3 as follows. Define $\mathbf{M} = -\frac{1}{L}\nabla^2 f(x_0) + \left(1 - \frac{3\delta}{4L}\right)\mathbf{I}$. We immediately notice that

- all eigenvalues of $\nabla^2 f(x_0)$ in $\left[\frac{-3\delta}{4}, L\right]$ are mapped to the eigenvalues of $\mathbf{M}$ in $[-1, 1]$, and

- any eigenvalue of $\nabla^2 f(x_0)$ smaller than $-\delta$ is mapped to eigenvalue of $\mathbf{M}$ greater than $1 + \frac{\delta}{4L}$.

Therefore, as long as $T \geq \widetilde{\Omega}\left(\frac{L}{\delta}\right)$, if we compute $x_{T+1} = x_0 + \mathbf{M}^T \xi$ for some random vector $\xi$, by the theory of power method, $x_{T+1} - x_0$ must be a negative-curvature direction of $\nabla^2 f(x_0)$ with value $\leq \frac{1}{2}\delta$. There are *two issues* with this approach.

The first issue is that, the degree $T$ of this matrix polynomial $\mathbf{M}^T$ can be reduced to $T = \widetilde{\Omega}\left(\frac{\sqrt{L}}{\sqrt{\delta}}\right)$ if the so-called Chebyshev polynomial is used.

    **Claim 4.1.** *Let $\mathcal{T}_t(x)$ be the $t$-th Chebyshev polynomial of the first kind, defined as:*

$$\mathcal{T}_0(x) := 1, \qquad \mathcal{T}_1(x) := x, \qquad \mathcal{T}_{n+1}(x) := 2x \cdot \mathcal{T}_n(x) - \mathcal{T}_{n-1}(x)$$

*then $\mathcal{T}_t(x)$ satisfies (see Trefethen [30]):*

$$\mathcal{T}_t(x) = \begin{cases} \cos(n \arccos(x)) \in [-1, 1] & \text{if } x \in [-1, 1]; \\ \frac{1}{2}\left[\left(x - \sqrt{x^2 - 1}\right)^n + \left(x + \sqrt{x^2 - 1}\right)^n\right] & \text{if } x > 1. \end{cases}$$

Since $\mathcal{T}_t(x)$ stays between $[-1, 1]$ when $x \in [-1, 1]$, and grows to $\approx (1 + \sqrt{x^2 - 1})^t$ for $x \geq 1$, we can use $\mathcal{T}_T(\mathbf{M})$ in replacement of $\mathbf{M}^T$. Then, any eigenvalue of $\mathbf{M}$ that is above $1 + \frac{\delta}{4L}$ shall grow

in a speed like $(1 + \sqrt{\delta/L})^T$, so it suffices to choose $T \geq \widetilde{\Omega}\left(\frac{\sqrt{L}}{\sqrt{\sigma}}\right)$. This is quadratically faster than applying the power method, so in $\texttt{Neon2}^{\mathsf{det}}$ we wish to compute $x_{t+1} \approx x_0 + \mathcal{T}_t\left(\mathbf{M}\right)\xi$.

The second issue is that, since we cannot compute Hessian-vector products, we have to use the gradient difference to approximate it; that is, we can only use $\mathcal{M}(y)$ to approximate $\mathbf{M}y$ where

$$\mathcal{M}(y) := -\frac{1}{L}\left(\nabla f(x_0 + y) - \nabla f(x_0)\right) + \left(1 - \frac{3\delta}{4L}\right) y \ .$$

How does error propagate if we compute $\mathcal{T}_t\left(\mathbf{M}\right)\xi$ by replacing $\mathbf{M}$ with $\mathcal{M}$? Note that this is a very non-trivial question, because the coefficients of the polynomial $\mathcal{T}_t(x)$ is as large as $2^{O(t)}$.

It turns out, the way that error propagates depends on how the Chebyshev polynomial is calculated. If the so-called backward recurrence formula is used, namely,

$$y_0 = 0, \quad y_1 = \xi, \quad y_t = 2\mathcal{M}(y_{t-1}) - y_{t-2}$$

and setting $x_{T+1} = x_0 + y_{T+1} - \mathcal{M}(y_T)$, then this $x_{T+1}$ is sufficiently close to the exact value $x_0 + \mathcal{T}_t\left(\mathbf{M}\right)\xi$. This is known as the stability theory of computing Chebyshev polynomials, and is proved in [6]. We defer all the proof details to Appendix C.2. $\qquad\square$

## 5  Neon2 in the Finite-Sum Setting

Let us recall how the shift-and-invert (SI) approach [26] solves the minimum eigenvector problem. Given matrix $\mathbf{A} = \nabla^2 f(x_0) \in \mathbb{R}^{d \times d}$ and suppose its eigenvalues are $-L \leq \lambda_1 \leq \cdots \leq \lambda_d \leq L$. At a high level, the SI approach

- chooses $\lambda = \delta - \lambda_1$ ,[5]
- defines positive definite matrix $\mathbf{B} = (\lambda \mathbf{I} + \mathbf{A})^{-1}$, and
- applies power method for a logarithmic number of rounds to $\mathbf{B}$ to find its approximate maximum eigenvector $v$.[6]

One can show that this unit vector $v$ satisfies $\lambda_1 \leq v^\top \mathbf{A} v \leq \lambda_1 + O(\delta)$ [13].

To apply power method to $\mathbf{B}$, one needs to compute matrix inversion $\mathbf{B}y = (\lambda \mathbf{I} + \mathbf{A})^{-1}y$ for arbitrary vectors $y \in \mathbb{R}^d$. The stability of SI ensures that it suffices to compute $\mathbf{B}y$ to some sufficiently high accuracy.[7]

One efficient way to compute $\mathbf{B}y$ to such high accuracy is by expressing $\mathbf{A}$ in a finite-sum form and then adopt convex optimization [13]. We call this approach *finite-sum SI*. Consider a *convex* quadratic function that is of a *finite sum of non-convex* functions:

$$g(z) := \frac{1}{2} z^\top (\lambda \mathbf{I} + \mathbf{A}) z + y^\top z = \frac{1}{n} \sum_{i=1}^n \left( \frac{1}{2} z^\top (\lambda \mathbf{I} + \nabla^2 f_i(x_0)) z + y^\top z \right) =: \frac{1}{n} \sum_{i=1}^n g_i(z) \ .$$

Now, computing $\mathbf{B}y$ is equivalent to minimizing $g(z)$, and one can use a stochastic first-order method to minimize it.

One such method is $\texttt{KatyushaX}$, which directly accelerates the so-called SVRG method using momentum, and finds $z$ using $\widetilde{O}(n + n^{3/4}\sqrt{L/\delta})$ computations of stochastic gradients.[8] Whenever a stochastic gradient $\nabla g_i(z) = (\lambda \mathbf{I} + \nabla^2 f_i(x_0))z + y$ is needed at some point $z \in \mathbb{R}^d$ for some random $i \in [n]$, instead of evaluating it exactly (which require a Hessian-vector product), we use $\nabla f_i(x_0 + z) - \nabla f_i(x_0)$ to approximate $\nabla^2 f_i(x_0) \cdot z$. We call this method $\texttt{Neon2}^{\mathsf{finite}}$.

Of course, one needs to show that KatyushaX is stable to noise. Using similar techniques as the previous two sections, one can show that the error term is proportional to $O(\|z\|_2^2)$, and thus as long as we bound the norm of $z$ is bounded (just like we did in the previous two sections), this should not affect the performance of the algorithm. We decide to ignore the detailed theoretical proof of this result, because it will complicate this paper.

**Theorem 5** (Neon2$^{\text{finite}}$). *Let $f(x) = \frac{1}{n}\sum_{i=1}^{n} f_i(x)$ where each $f_i$ is $L$-smooth and $L_2$-second-order smooth. For every point $x_0 \in \mathbb{R}^d$, every $\delta > 0$, every $p \in (0,1]$, the output $v = \text{Neon2}^{\text{finite}}(f, x_0, \delta, p)$ satisfies that, with probability at least $1 - p$:*

1. *If $v = \bot$, then $\nabla^2 f(x_0) \succeq -\delta\mathbf{I}$.*

2. *If $v \neq \bot$, then $\|v\|_2 = 1$ and $v^\top \nabla^2 f(x_0) v \leq -\frac{\delta}{2}$.*

*Moreover, the total number stochastic gradient evaluations is $\widetilde{O}\big(n + \frac{n^{3/4}\sqrt{L}}{\sqrt{\delta}}\big)$, where the $\widetilde{O}$ notion hides logarithmic factors in $d, 1/p$ and $L/\delta$.*

# 6 Applications of Neon2

We show how Neon2 can be applied to existing algorithms such as SGD, GD, SCSG, SVRG, Natasha2, CDHS. Unfortunately, we are unaware of a generic statement for applying Neon2 to *any* algorithm. Therefore, we have to prove them individually.[9]

Throughout this section, we assume that some starting vector $x_0 \in \mathbb{R}^d$ and upper bound $\Delta_f$ is given to the algorithm, and it satisfies $f(x_0) - \min_x\{f(x)\} \leq \Delta_f$. This is only for the purpose of proving theoretical bounds. Since $\Delta_f$ only appears in specifying the number of iterations, in practice, one can run enough number of iterations and then halt the algorithm, without knowing $\Delta_f$.

## 6.1 Applying Neon2 to SGD and GD

To apply Neon2 to turn SGD into an algorithm finding approximate local minima, we propose the following process Neon2+SGD (see Algorithm 4). In each iteration $t$, it first applies SGD with mini-batch size $O(\frac{1}{\varepsilon^2})$ (see Line 4). Then, if SGD finds a point with small gradient, we apply Neon2$^{\text{online}}$ to decide if it has a negative curvature, if so, then we move in the direction of the negative curvature (see Line 10). We have the following theorem:

**Theorem 7a.** *With probability at least $1 - p$, Neon2+SGD outputs an $(\varepsilon, \delta)$-approximate local minimum in gradient complexity $T = \widetilde{O}\Big(\big(\frac{\mathcal{V}}{\varepsilon^2} + 1\big)\big(\frac{L_2^2\Delta_f}{\delta^3} + \frac{L\Delta_f}{\varepsilon^2}\big) + \frac{L^2}{\delta^2}\frac{L_2^2\Delta_f}{\delta^3}\Big)$.*

**Corollary 6.1.** *Treating $\Delta_f, \mathcal{V}, L, L_2$ as constants, we have $T = \widetilde{O}\big(\frac{1}{\varepsilon^4} + \frac{1}{\varepsilon^2\delta^3} + \frac{1}{\delta^5}\big)$.*

One can similarly (and more easily) give an algorithm Neon2+GD, which is the same as Neon2+SGD except that the mini-batch SGD is replaced with a full gradient descent, and the use of Neon2$^{\text{online}}$ is replaced with Neon2$^{\text{det}}$. We have the following theorem:

**Theorem 7c.** *With probability at least $1 - p$, Neon2+GD outputs an $(\varepsilon, \delta)$-approximate local minimum using $\widetilde{O}\Big(\frac{L\Delta_f}{\varepsilon^2} + \frac{L^{1/2}}{\delta^{1/2}}\frac{L_2^2\Delta_f}{\delta^3}\Big)$ full gradient computations.*

We only prove Theorem 7a in Appendix D and the proof of Theorem 7c is only simpler.

## 6.2 Other Applications

Due to space limitation, we defer the applications to Natasha2, CDHS, and SCSG to Appendix A. At a high level, the applications to Natasha2 and CDHS are trivial because NC-search was already a subroutine required by both algorithms, so one can directly replace them with Neon2 of this paper. The application to SCSG is less non-trivial, because one has to additionally take care of some probabilistic behavior from SCSG.

# Acknowledgements

We would like to thank Tianbao Yang and Yi Xu for helpful feedbacks on this manuscript. This work was done when Yuanzhi Li was a summer intern at Microsoft Research in 2017.

## Footnotes

[4]The original paper of CDHS only stated their algorithm in the deterministic setting, but is easily verifiable to work in the finite-sum setting, see discussions in [1].

[5]The precise SI approach needs to binary search $\lambda$ because $\lambda_1$ is unknown.

[6]More precisely, applying power method for $O(\log(d/p))$ rounds, one can find a unit vector $v$ such that $v^\top \mathbf{B} v \geq \frac{9}{10}\lambda_{\max}(\mathbf{B})$ with probability at least $1 - p$. One can also prove that this vector $v$ satisfies $\lambda_1 \leq v^\top \mathbf{A} v \leq \lambda_1 + O(\delta)$.

[7]More precisely, if suffices to compute $w \in \mathbb{R}^d$ so that $\|w - \mathbf{B}y\| \leq \varepsilon\|y\|$, in a time complexity that polynomially depends on $\log \frac{1}{\varepsilon}$ [5, 13].

[8]Shalev-Shwartz [29] first discovered that one can apply SVRG to minimize sum-of-nonconvex functions. It was also observed that applying APPA/Catalyst reductions to SVRG one can achieve accelerated convergence rates [13, 29], and this approach is commonly known as AccSVRG. However, AccSVRG requires some careful parameter tuning of its inner loops, and thus is a logarithmic-factor slower than $\texttt{KatyushaX}$ and also less practical [3].

[9]This is because stationary-point finding algorithms have somewhat different guarantees. For instance, in mini-batch SGD we have $f(x_t) - \mathbb{E}[f(x_{t+1})] \geq \Omega(\|\nabla f(x_t)\|^2)$ but in SCSG we have $f(x_t) - \mathbb{E}[f(x_{t+1})] \geq \Omega(\mathbb{E}[\|\nabla f(x_{t+1})\|^2])$.

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
