[Reviews · NeurIPS 2018]

Reviewer 1



The paper proposes adaptations of the existing NC (Negative Curvature) search algorithm called "Neon2". Neon2 follows an iterative process and uses stochastic updates to solve the NC (Negative Curvature) search problem. The authors provide a discussion of their Neon2 variants for each of these cases and discuss improved theoretical bounds for them. I like the fact that Neon2 can be used as a black-box in applications where NC is applicable. Such applications fall into two broad settings - online and offline, and the authors have provided algorithms and discussion in detail for both these cases. This is useful in understanding the context where Neon2 is relevant. The authors have delved deeper into the theoretical guarantees and that is a plus of this work. However, I feel the organization of the paper could be improved. Readability of the paper is poor. There are several versions of the methods introduced in a haphazard fashion and I feel it doesn't flow well. For eg, in line 127 the authors suddenly mention Neon2_online_weak, at which point I am lost as to that "weak" means. line: 63-64: "In the end, the vector xT − x0 gives us enough information about the negative curvature." >> needs to be explained more clearly why. was not clear to me why this is the case. - Even though this work is a theoretical contribution, I am also curious about how the NC adaptation as Neon2 works empirically. Fig 1 is reference from the proofs, the placement of the Figure on second page is confusing. Also, it the figure needs to be explained better. line 32: Figure 1 >> legends confusing, needs to be made more clear. The authors acknowledge that a parallel work on similar lines as their appeared few days before them called "Neon". Although they claim "Neon2" is different from "Neon" and has better theoretical properties, this should be discussed in more detail in the related work section. Empirical comparison would help too.

Reviewer 2



This paper proposes an efficient negative-curvature search approach, Neon2, which can fast compute a direction to reduce the objective when the gradient is small. Neon2 has two versions, one is for online setting and another is for deterministic setting. The main function of Neon2 is to escape saddle points. It can convert existing stationary point finding algorithms, including SGD, SCGC, Natasha2, GD and CDHS, into local minimum finding algorithms. In particular, Neon2 turns Natasha2 into a first-order method without hurting its computational complexity. When combining with Neon2, the complexity of most algorithms usually outperforming best known results. On the other hand, Neon2 outperforms its counterpart Neon which is also designed for negative-curvature search in both online setting and deterministic setting. So this guarantees that the algorithm taking Neon2 to escape saddle point is better than the algorithm which is combined with Neon. I also read the proofs and cannot find any bugs. So I think that this paper makes great contributions to optimization area, especially for escaping saddle points. The weakness is that there is no any comparison experiments to illustrate the effectiveness of escaping saddle points and higher efficiency over other methods, such as Neon. Although I note that most of escaping saddle point algorithms have no experiments, it is better to run several experiments or explain why it is hard to design the experiments. It seems to be hard to run this algorithm in practice. This is because for the proposed Neon2, its parameters, such as the sufficient large constant C0, the norm of Gaussian noise \sigma, learning rate eta, are selected very carefully such that it can give good theoretical results, while some key parameters, e.g. the first and second-order -Lipschitz smooth parameter L and L2, are unknown. So how to determine these parameters? As mentioned above, since there is no experiment, it is a little hard for reader to implement the algorithms for escaping saddle points.

Reviewer 3



This paper provides a way to turn algorithms that find a first-order stationary point (i.e. gradient norm is bounded by a constant) to algorithms that find a second-order stationary point (a.k.a. approximate local minima). The definition of a second order stationary point is that the gradient norm at the point is bounded by a constant AND the smallest eigenvalue of the Hessian is at least a negative number (but with small magnitude). The main contribution here is that the proposed approach avoids Hessian or Hessian-vector product computations. It just needs (stochastic) gradient information, which is a significant contribution. Previous works need Hessian-vector products for computing the eigenvector that corresponds to the most negative eigenvalue of the Hessian; it is recognized that the negative curvature can be exploited to decrease the objective value even when the gradient norm is small. In this paper the authors provide two methods to replace the Hessian-vector product computations. One for the offline setting, which is that setting that the convergence rate depends on number of samples n. The other for the online setting, which is the setting that the convergence rate does not depend on $n$, as $n$ can be infinite. Theoretical guarantees on both offline setting and online setting are provided. I enjoy reading the paper. The analysis looks correct and sound, though I didn't have time to read them in details. One concern is that the complexity result in this paper may not be the state-of-the-art. "SPIDER: Near-Optimal Non-Convex Optimization via Stochastic Path Integrated Differential Estimator" might have a better rate. Yet, I understand that the paper is put on arxiv after this submission. It is unfair to compare them. Still, I suggest the authors to briefly mention it in the next version. (p.s. It seems like SPIDER has a state-of-the-art rate in finding a first order stationary point. The authors of SPIDER follow the nice tricks proposed in this paper to exploit the negative curvature and turn it to an approximate local-minimum finding one. So, the papers are actually complementary.)